# Advances in Soybean Genetic Improvement

**DOI:** 10.3390/plants13213073

**Published:** 2024-10-31

**Authors:** Adriana Vargas-Almendra, Roberto Ruiz-Medrano, Leandro Alberto Núñez-Muñoz, José Abrahán Ramírez-Pool, Berenice Calderón-Pérez, Beatriz Xoconostle-Cázares

**Affiliations:** 1Departamento de Biotecnología y Bioingeniería, Centro de Investigación y de Estudios Avanzados (CINVESTAV), Ciudad de México 07360, Mexico; adriana.vargas@cinvestav.mx (A.V.-A.); rmedrano@cinvestav.mx (R.R.-M.); leandro.nunez@cinvestav.mx (L.A.N.-M.); jramirezp@cinvestav.mx (J.A.R.-P.); bcalderon@cinvestav.mx (B.C.-P.); 2Programa de Doctorado Transdisciplinario en Desarrollo Científico y Tecnológico para la Sociedad, Centro de Investigación y de Estudios Avanzados (CINVESTAV), Av. Instituto Politécnico Nacional 2508, Ciudad de México 07360, Mexico

**Keywords:** soybean, genetic transformation, crop improvement, biotic stress, abiotic stress

## Abstract

The soybean (*Glycine max*) is a globally important crop due to its high protein and oil content, which serves as a key resource for human and animal nutrition, as well as bioenergy production. This review assesses recent advancements in soybean genetic improvement by conducting an extensive literature analysis focusing on enhancing resistance to biotic and abiotic stresses, improving nutritional profiles, and optimizing yield. We also describe the progress in breeding techniques, including traditional approaches, marker-assisted selection, and biotechnological innovations such as genetic engineering and genome editing. The development of transgenic soybean cultivars through *Agrobacterium*-mediated transformation and biolistic methods aims to introduce traits such as herbicide resistance, pest tolerance, and improved oil composition. However, challenges remain, particularly with respect to genotype recalcitrance to transformation, plant regeneration, and regulatory hurdles. In addition, we examined how wild soybean germplasm and polyploidy contribute to expanding genetic diversity as well as the influence of epigenetic processes and microbiome on stress tolerance. These genetic innovations are crucial for addressing the increasing global demand for soybeans, while mitigating the effects of climate change and environmental stressors. The integration of molecular breeding strategies with sustainable agricultural practices offers a pathway for developing more resilient and productive soybean varieties, thereby contributing to global food security and agricultural sustainability.

## 1. Introduction

The soybean (*Glycine max* (L.) Merr.) originated in China and has become globally important as a source of nutrition for humans and animals, as well as for bioenergy production [1]. The species has 40 chromosomes (2n = 40), and its reproductive system is predominantly self-pollinating, with cleistogamous flowers, leading to a 97–99% self-pollination rate [2,3]. The soybean’s versatile applications have positioned it as a critical component in global agronomic and trade systems. In 2020, the USA, Brazil, and Argentina contributed to 80% of global soybean production with an estimated of 338 million metric tons [4,5,6]. As a primary source of protein and oil, the soybean is integral to the agricultural sector, contributing to economic stability and promoting sustainable practices, particularly through crop rotation systems that enhance soil fertility [7,8,9].

The soybean contributes 25% of the global commercial vegetable oil market and 70% of available plant-based protein source [10]. Its oil is fundamental for human consumption, biodiesel, lubricants, and bio-based polymers production, containing 13% saturated fats and unsaturated fatty acids like oleic, linoleic, and linolenic acids [11,12]. With 35% protein content (on a 13% moisture basis), the soybean surpasses other staple crops like wheat, rice, and corn, and is rich in antioxidants, minerals, vitamins, and dietary fiber [13,14]. Additionally, soybeans contain bioactive compounds, such as isoflavones and saponins, which are linked to health benefits, including potential menopause delay and cancer prevention [4,5]. Soybean products, especially those from fermentation (including soy protein, soy sauce, soy milk and soybean products), are extensively used in Asia for human consumption and are vital in the livestock and poultry industries for high-quality feed [13]. Its nutritional profile and high protein content make it a key component of balanced diets, addressing malnutrition and supporting vegetarian diets, thus promoting global food security and human welfare [15].

Soybean cultivation is most successful in warm and moist climates, with one or two agricultural cycles depending on water availability. The cultivation process begins with land preparation, followed by mechanical or manual sowing, often using fungicide-treated seeds to prevent rotting. Phosphorus and potassium applications are required [16]. Inoculation with *Rhizobium* is recommended to enhance nitrogen fixation and maintaining soil pH between 6.0 and 7.5 is crucial to avoid aluminum toxicity [17]. Many commercial soybean cultivars are sensitive to photoperiod, with flowering occurring under short-day conditions. Soybean plants typically take 75–80 days to reach full maturity, depending on day length exposure. However, anti-nutritional factors such as protease inhibitors, lectins, phytates and tannins, can impair nutrient assimilation [18]. Despite its benefits, soybean cultivation faces challenges related to sustainable production, land-use dynamics, and the need for genetic improvements [19].

This review addresses the impact of biotic stressors on soybean cultivation, highlighting major pests, diseases, and associated management strategies. Challenges posed by abiotic stress, including drought, salinity, and extreme temperatures, are examined alongside the molecular mechanisms and breeding strategies developed to mitigate these effects. Subsequent sections focus on conventional breeding, genetic engineering, and genome-editing approaches that have revolutionized the soybean genetic improvement. Finally, advances in soybean transformation techniques are discussed, along with future directions in soybean research and the role of biotechnology in promoting sustainable agriculture.

## 2. Biotic Stress in Soybean Cultivars

Biotic stress in soybean crops is driven by a wide diversity of pests and pathogens, varying by geography and climate (Figure 1A,B). Bacterial infections, such as those caused by *Pseudomonas syringae* and *Xanthomonas campestris*, produce small brown pustules and blight spots, especially under wet and cool conditions, although their impact on yield is generally minor [20]. In contrast, fungal pathogens pose a significant threat to its productivity, with *Phakopsora pachyrhizi* being particularly devastating, causing premature defoliation and reduced photosynthesis [21]. Other fungal diseases in soybeans include *Septoria glycines*, *Diaporthe phaseolorum*, *Fusarium* spp., *Cercospora kikuchii*, *Cercospora sojina*, *Colletotrichum truncatum*, *Corynespora cassiicola*, Glomerella glycines, *Macrophomina phaseolina*, *Peronospora manshurica*, *Phakopsora meibomiae*, *Phomopsis* spp., *Sclerotium rolfsii*, *Sclerotinia sclerotiorum*, and *Rhizoctonia solani* infections, all of which impair plant health [20,21].

Viral infections pose a significant threat to soybean production, leading to reduced plant vigor, impaired photosynthetic efficiency, and substantial yield losses [22]. A variety of viruses have been documented to infect soybean plants, including Alfamovirus AMV, *Begomovirus vignaradiatae*, *Comovirus siliquae*, *Ilavirus* spp., *Ilarvirus* TSV, *Luteovirus glycinis*, *Nepovirus nicotianae*, *Orthotospovirus glycininecrovenae*, *Potyvirus glycitessellati*, *Potyvirus phaseoluteum*, *Potyvirus phaseovulgaris*, *Potyvirus trifolii*, *Potyvirus vignae*, and Sobemovirus SBMV [23]. In addition, other biotic stressors such as nematodes—specifically *Belonolaimus longicaudatus*, *Heterodera glycines*, *Meloidogyne* spp., and *Rotylenchulus reniformis* (Figure 1B), and oomycetes, notably *Phytophthora sojae*—are also important pathogens that adversely affect soybean health and productivity [24,25].

Insect pests further compromise soybean productivity by causing direct damage to crops and facilitating the spread of plant pathogens. In Asia, primary insect threats include *Acalymma vittata*, *Diabrotica undecimpunctata*, *Spodoptera exigua*, and *Spodoptera praefica*, with climate change accelerating their evolution [26]. In the Americas, major pests are *Anticarsia gemmatalis*, *Aphis glycines*, *Bemisia tabaci*, *Chrysodeixis includens*, and *Sternechus subsignatus*, which damage crops through defoliation, nutrient depletion, and virus transmission. Other pests that can affect soybean cultivation include *Ceratoma trifurcata*, *Chinavia hilaris*, *Euschistus* spp., *Halyomorpha halys*, *Helicoverpa zea*, *Hypena scabra*, *Megacopta cribraria*, *Nezara viridula*, *Piezodorus guildinii*, *Spissistilus festinus*, and *Tetranychus urticae* [27,28].

Weeds are a major factor contributing to yield losses in soybean production due to competition for essential resources such as space, water, light, and nutrients. Additionally, weeds can exert negative effects through allelopathy and by serving as hosts for pests and diseases. The impact of weed competition is influenced by weed density, species composition, the competitive ability of the soybean cultivar, soil conditions, crop management practices, and the duration of coexistence between the crop and weeds. Weed interference can result in grain yield reductions by up to 80%. Common weed species that compete with the soybean include *Amaranthus dubius*, *Brachiaria platyphylla*, *Caperonia palustre*, *Cassia tora*, *Commelina elegans*, *Cucumis* spp., *Cynodon dactylon*, *Cyperus rotundus*, *Echinochloa colonum*, *Eleusine indica*, *Ipomoea tiliacea*, *Kallstroemia maxima*, *Milleria quinqueflora*, *Parthenium hysterophorus*, *Portulaca oleracea*, and *Sorghum halepense* [29]. Interestingly, the soybean can be used in agricultural systems, for example, intercropping with maize has been showed to protect against infestation by the root parasitic plant *Striga* spp., which underscores another application of soybean cultivation [30].

The complex interactions among biotic stressors demand the development of comprehensive management strategies [31]. The rise of resistant pest populations and virulent pathogen strains, exacerbated by global trade and intensive agricultural practices, highlights the need for continuous monitoring, stringent biosecurity measures, and integrated pest management (IPM) approaches. Effective IPM strategies include the deployment of pest-resistant cultivars, implementation of cultural control practices, utilization of biological control agents, crop rotation, and sanitation measures (Figure 1C). These approaches aim to reduce reliance on chemical interventions and promote sustainable agricultural systems [32].

## 3. Abiotic Stress in Soybean Cultivars

Abiotic stress poses a substantial threat to soybean productivity by affecting growth, development, yield, and key physiological processes such as photosynthesis, water uptake, and nutrient absorption [33]. Recent research has increasingly focused on elucidating the complex responses of the soybean to abiotic stressors, including drought, salinity, extreme temperatures, soil pH imbalances, and nutrient deficiencies [34]. Understanding these dynamics is crucial for developing resilient cultivars and optimizing management practices to mitigate the effects of these stressors on soybean production.

For example, drought stress leads to reduced water availability, triggering stomatal closure and hampering carbon assimilation, thereby impeding overall plant growth [35]. Drought, particularly during seed germination and flowering, can reduce yields by over 50%, leading to substantial economic losses [36,37]. Drought-tolerant soybean cultivars employ avoidance and tolerance mechanisms as well as structural and biochemical adaptations (such as wax and trichome production on leaves, stomatal opening regulation, reduction of photosynthetic rate and the synthesis of osmoregulatory compounds) to mitigate water deficit [38,39,40]. The abscisic acid (ABA)-dependent and independent pathways are crucial for controlling water allocation and photosynthesis during stress [41,42]. The ABA is synthesized in roots and likely in the leaf vascular system; it has been postulated that ABA is transported from the roots to leaves, and in general to aerial tissues, through the xylem and translocated where it induces stomatal closure [43]. However, there is evidence that ABA impacts more stomatal opening than the xylem, at least in tomatoes (*Solanum lycopersicum*) [44]. Consistent with this, ABA in soybeans appears to accumulate first in shoots where it is transported to roots, presumably through the phloem, where adaptive responses to water deficit occur [45]. Recent work supports the notion that ABA synthesized in guard cells has a short-term effect on stomatal regulation while ABA originating from vasculature has a role in long-term responses in roots to water availability [46]. Additionally, responses to water deficits and flooding studied in soybeans with contrasting phenotypes indicate that tolerant cultivars accumulate higher ABA content in leaves than susceptible ones, and show an increased number of plastoglobules, which are lipoprotein bodies bound to thylakoids that accumulate antioxidants and other molecules that may help to cope with reactive oxygen species produced during drought stress [34,47,48]. Also, tolerant cultivars maintain high seed production as well as flower number and pollen germination [35]. Additionally, oxidative stress resulting from water scarcity is countered by the expression of osmoregulators and enzymes like catalase and superoxide dismutase in tolerant cultivars [49,50,51,52]. Transcriptomic studies have identified key drought-responsive genes, including *GmNAC*, *MYB*, *WRKY*, *AREB*, and *DREB*, overexpressed in tolerant germplasm and suggesting a coordinated, supracellular response that involves long-distance signaling [53,54,55,56,57,58,59,60,61]. For instance, the *WRKY* genes *GmWRKY12* and *GmWRKY54* mediate drought tolerance through ABA and Ca^+2^ signaling pathways, enhancing proline accumulation and drought resilience [58,62]. Some mRNAs induced by water deficit in a common bean (*Phaseolus vulgaris*) cultivar accumulate in the phloem, suggesting that such tolerance involves long-distance signaling through the vasculature, and may be a more general phenomenon [39].

The soybean is also sensitive to salinity, which affects seed quality, growth, and nodulation due to oxidative stress, ionic toxicity, and osmotic imbalance [63,64]. However, certain wild soybean accessions exhibit salinity tolerance [65]. Extreme temperatures further challenge soybean cultivation, impairing reproductive development [66].

### 3.1. Molecular Mechanisms and Signaling Pathways to Cope with Abiotic Stress

Under abiotic stress, plants activate various molecular and biochemical responses, including the expression of stress-responsive genes such as transcription factors (TFs), osmoprotectants, and antioxidant enzymes, which are crucial for enhancing stress tolerance [67]. Phytohormone signaling pathways, particularly those involving ABA, ethylene, and jasmonic acid, orchestrate adaptive responses by regulating stomatal closure, osmotic adjustments, and stress-related gene expression [68]. Understanding these intricate molecular networks is critical for developing strategies to bolster soybean resilience (Figure 2).

### 3.2. Impact of Abiotic Stress on Crop Productivity and Sustainability

Abiotic stress represents a significant challenge to global soybean cultivation, where climate exacerbates threats such as temperature fluctuations, heatwaves, droughts, floods, and UV radiation [25]. These stressors currently contribute to yield losses exceeding 40%, and in some cases reaching 80% depending on crop stage and environmental intensity [69]. Under optimal conditions, soybean yields range from 7.2 to 11 tons per hectare, but the global field average is significantly lower (2.3 to 3.4 tons), and stress conditions can reduce yields to less than 2 tons per hectare [70,71].

Additionally, abiotic stress accelerates disease development and the emergence of new pathogens, as evidenced by the 2012 drought in major soybean-producing countries, which increased *Macrophomina phaseolina* outbreaks, leading to unprecedented economic losses [72,73]. Addressing these challenges requires a comprehensive approach, integrating advanced molecular techniques, breeding strategies, and sustainable agronomic practices to enhance soybean resilience, sustain crop productivity, and support global food security [74]. Continued collaboration among researchers, agronomists, and policymakers will be essential in developing stress-resistant soybean cultivars capable of withstanding evolving environmental conditions.

## 4. Genetic Improvement of Soybeans

Efforts to improve soybean resilience to abiotic stress involve a multifaceted approach that integrates both conventional breeding and biotechnological interventions. Key strategies focus on breeding for traits such as improved water use efficiency, salt tolerance, and heat stress resilience [75]. Additionally, implementing agronomic practices, including precision farming, optimized irrigation management, and soil nutrient enhancement, further mitigates the adverse effects of stress on soybean productivity [76].

The development of soybean varieties with enhanced resistance to biotic and abiotic stresses offers substantial advantages to producers, reducing reliance on chemical inputs such as pesticides and fertilizers [67]. By incorporating traits such as drought tolerance, pest resistance, and herbicide tolerance, soybean producers can significantly reduce production costs and environmental impacts, supporting sustainable agricultural practices [77,78,79].

On the other hand, the presence of bioactive compounds such as isoflavones, tocopherols, and phytosterols provides natural antioxidant properties and potential health benefits, including cholesterol reduction, blood sugar regulation, and mitigation of cardiovascular diseases [80]. The high fiber content in soybeans further supports digestive health by promoting a balanced gut microbiota and preventing gastrointestinal disorders [81]. These improvements not only make soybeans a more valuable source of plant-based protein but also increase digestibility and reduce allergenic potential, thereby expanding their market acceptance and contributing to global food security. In that sense, improving the nutritional quality of soybean seeds by enhancing protein content, altering fatty acid profiles, and reducing anti-nutritional factors like phytic acid and protease inhibitors has direct implications for human health.

To achieve these advancements in soybean resilience and nutritional quality, a range of genetic improvement techniques have been employed, combining both traditional and modern biotechnological approaches. In the following sections, we explore key methodologies such as conventional breeding, marker-assisted selection, genetic engineering, and genome editing, all of which play a critical role in enhancing soybean traits and meeting the growing demands for sustainable agriculture.

### 4.1. Conventional Breeding

This approach employs diverse germplasms to incorporate desirable traits into progeny. The global soybean germplasm collection, comprising approximately 200,000 accessions conserved in over 70 countries—of which 30% are unique—serves as an important resource for breeding [82,83]. The soybean germplasm can be classified into the following three evolutionary types: wild relatives, local domesticates (landraces), and modern commercial cultivars. Despite the extensive diversity within this collection, only a small fraction has been used in breeding programs, resulting in elite cultivars with limited genetic diversity [83].

Incorporating wild soybean germplasm for the improvement of commercial cultivars presents an opportunity to enhance the genetic diversity that was reduced during domestication approximately 5000 years ago. This strategy would enable the development of soybean cultivars with resistance to the soybean cyst nematode (*H. glycines*), soybean aphid (*Aphis glycines*), and foxglove aphid (*Aulacorthum solani*), as well as for introducing traits like flowering regulation, photoperiod sensitivity, and drought and salinity tolerance. These traits are found in wild relatives and plant introductions of the *Glycine* species, including *G. soja*, *G. tomentella*, *G. tabacina*, and *G. latifolia* [84].

In conventional breeding, the utilization of male sterility systems can facilitate the development of hybrid soybean cultivars. Plant male sterility refers to the condition where the stamen develops abnormally, resulting in the inability to produce functional male gametes. Hybrid breeding, although traditionally challenging owing to its self-pollinating nature, can exploit heterosis (hybrid vigor), potentially leading to yield improvements. Male sterility systems, such as genic and cytoplasmic male sterility, enable the production of hybrids by ensuring cross-pollination [85,86]. Male sterility systems, such as genic and cytoplasmic male sterility, enable the production of hybrids ensuring cross-pollination [85,86]. Recent advances have focused on the identification of genes and molecular pathways governing male sterility, offering novel tools for hybrid soybean breeding.

### 4.2. Marker-Assisted Selection (MAS)

The association of desirable genotypes with specific genetic markers can accelerate the development of high-performance soybean cultivars. Initially, quantitative trait loci (QTL) linked to important traits such as protein and oil content were identified. This led to the discovery of 25 single nucleotide polymorphism (SNPs) associated with seed oil content and seven SNPs linked to both protein and oil content [87,88]. The advent of genome sequencing and annotation in various soybean genotypes has further accelerated genetic improvement through the identification of numerous SNPs. Earlier molecular techniques, such as restriction fragment length polymorphism (RFLP), enabled the construction of the first genetic map of the soybean genome [89]. Additionally, randomly amplified polymorphic DNA (RAPD), amplified fragment length polymorphism (AFLP), and simple sequence repeats (SSRs) have provided critical insights into the genetic diversity of the soybean [90,91,92]. Comprehensive analysis of commercial soybean plant introductions has identified SNPs as key elements for improving traits such as protein and oil content, as well as enhancing tolerance to biotic and abiotic stress [93]. This approach has been instrumental in identifying genetic variation in coding and non-coding regions, thereby facilitating the development of elite soybean cultivars.

Additional tools such as genomic selection (GS) represent an emerging alternative to traditional MAS, demonstrating enhanced efficiency in improving complex quantitative traits in soybeans. Whereas MAS focuses on identifying and selecting QTL, GS employs genome-wide marker data to predict the performance of individuals based on their genetic composition [94]. This approach proves advantageous for traits controlled by numerous genes with small effects, which are challenging to improve using MAS alone. The GS analyses genetic and phenotypic data from a training set (TS) to estimate the genomic estimated breeding values of individuals in a prediction set (PS). This methodology allows breeders to make selection decisions earlier in the breeding cycle, without full phenotypic data. Studies have demonstrated that GS can achieve high predictive accuracy, particularly when the TS is large and genetically related to the PS, leveraging similarities in linkage disequilibrium [95]. In soybeans, GS has achieved prediction accuracies of 0.5 to 0.7 for yield when utilizing a TS composed of full siblings [96]. By integrating GS into breeding programs, the selection of complex traits can be expedited, thereby increasing genetic gain and enhancing the efficiency of breeding programs. This program renders GS a potent complement to MAS, enabling breeders to address the limitations of traditional marker-assisted approaches to traits with polygenic inheritance.

### 4.3. Genetic Engineering

Recombinant DNA technologies have enabled the rapid and precise development of new soybean cultivars, surpassing traditional breeding methods. The main advantage of genetic engineering is the possibility to introduce foreign genes to confer novel traits on crops in a very specific manner. The methods used are through the insertion of the foreign gene through *Agrobacterium*-mediated transformation, and by biolistic delivery of metal microparticles coated with the foreign DNA to be introduced [97]. The first genetically modified soybean cultivars were generated for tolerance to the herbicide glyphosate and insect pests by introducing genes for proteins from *Agrobacterium* spp. strain CP4 and *Bacillus thuringiensis*, respectively [98]. In addition, herbicide resistance remains the primary focus of genetic engineering in soybeans for commercialization purposes, given its significant role in facilitating large-scale agricultural practices. These cultivars allowed the expansion of soybean cultivation areas in the late 1990s, and, by 2017, transgenic cultivars accounted for 80% of global soybean production [99]. Current genetic engineering efforts are focused on the following three main goals: (1) enhancing crop traits like protein and oleic acid content, (2) increasing yield, and (3) improving resistance to biotic and abiotic stresses [78]. Despite the potential benefits, the widespread adoption of genetically modified soybean cultivars is constrained by limited public acceptance and stringent regulatory frameworks. There are also technical hurdles that need to be solved for the efficient genetic transformation of the most relevant crops, including soybeans, one of them being the recalcitrance of this and other species to in vitro plant regeneration. This issue limits the use of this technique to certain genotypes which may not necessarily be optimal in terms of desired traits such as protein and oil content as well as tolerance to biotic and abiotic stress. Transformation of hairy roots as well as introducing enhancers of regeneration such as the WUS gene may help to solve this issue [97,100].

### 4.4. Genome Editing

This strategy has revolutionized crop improvement by enabling precise modifications at specific loci within the plant genome. Techniques such as CRISPR/Cas9 are noteworthy for the ability to introduce targeted changes, including insertions/deletions (indels) or replacements, enhancing desirable traits in crops like soybeans. Genome editing allows for the rapid development of new soybean cultivars by directly targeting genes responsible for critical agricultural traits. Other technologies have been applied to soybean, including CRISPR/Cpf1, CRISPR/LbCpf1, CRISPR/Lbcas12-a, CRISPR/Cas12a, and cytosine base editors, among others [101,102,103,104]. Particularly in soybeans, genome editing tools have been used to confer disease resistance, enhance drought and salinity tolerance, reduce phytic acid content, modify flowering time, improve yield, facilitate functional genomic studies, and enhance herbicide tolerance, plant architecture, oil composition, and palatability (Table 1). Unlike traditional genetic engineering, genome editing does not necessarily involve the introduction of foreign DNA, which can increase its acceptance among consumers and regulatory bodies due to fewer concerns about the introduction of transgenic elements [78]. However, similar barriers exist with this technique, as plant regeneration is a limiting step [97,100].

## 5. Overview of Soybean Genetic Transformation

Genetic transformation is a biotechnological approach enabling modification of the plant genome to incorporate desirable traits [75,161]. This technology has revolutionized soybean crop improvement by reducing reliance on chemical inputs and promoting sustainable agricultural practices. It holds significant potential for enhancing agricultural productivity, food security, and environmental sustainability. Soybean genetic transformation begins with the identification and selection of target traits for enhancement, such as herbicide tolerance, insect resistance, or improved nutritional quality (Figure 3A). This is followed by the isolation and manipulation of specific genes or genomic regions related to the desired traits [162]. The selected DNA sequences are introduced into the soybean genome using methods such as *Agrobacterium*-mediated transformation (Figure 3B) or biolistic particle delivery systems (Figure 3C) [163]. Successful integration of the recombinant DNA results in transgenic soybean plants exhibiting the desired traits (Figure 3D).

### 5.1. Agrobacterium-Mediated Transformation

*Agrobacterium tumefaciens* is a plant pathogenic bacterium widely used in genetic engineering due to its ability to transfer a segment of its DNA, known as T-DNA, into plant genomes, resulting in the formation of galls or tumors [164]. This capability has been harnessed for the development of transgenic plants by replacing the tumor-inducing genes within the bacterial T-DNA with genes of interest. The transformation process generally involves the following three stages: cloning the gene of interest into a binary vector, co-culturing the genetically engineered *Agrobacterium* with plant tissues to facilitate the infection and subsequent T-DNA transfer, and selecting and regenerating transformed explants using plant tissue culture techniques [162]. Although *Agrobacterium*-mediated transformation can be relatively slow and genotype-dependent, it is a cost-effective method that enables stable transgene integration [165,166].

Recently, *Ochrobactrum haywardense* H1-8 has been identified as an alternative for soybean transformation, providing higher efficiency and marker-free transformation compared to *Agrobacterium* strains [167].

Genetic transformation of the soybean, a recalcitrant species, remains challenging due to low transformation and regeneration efficiencies [168]. Recent efforts have focused on optimizing protocols across various cultivars through different techniques, explants, and culture conditions. For instance, in *Agrobacterium*-mediated transformation, the selection of the explant is critical as it directly influences plant–*Agrobacterium* interaction and the overall transformation success. Two primary methods are employed, the cotyledonary node and the half-seed methods, the latter using the embryonic axis attached to one cotyledon [169,170]. Optimization strategies have included the selection of *A. tumefaciens* strains, adjustments in co-culture conditions, and the development of methods to reduce tissue oxidation (Table 2). Antioxidants and other chemical components have been tested to reduce oxidative stress and enhance the virulence of the bacterial strain [171].

Despite significant advancements, challenges remain in regulatory frameworks, public acceptance, and ecological considerations. Effective collaboration between researchers and policymakers is essential to develop comprehensive regulations and foster public understanding of genetically modified soybeans [172]. Ongoing refinement of genetic modification techniques and in-depth exploration of the soybean genome using advanced molecular tools are essential for future improvements.

**Table 2 plants-13-03073-t002:** Protocols for soybean *Agrobacterium*-mediated transformation.

Soybean Cultivar	Method	*A. tumefaciens* Strain	Co-Culture			Remarks	TrF (%)	Ref.
			T (°C)	Pp	Time (days)			
A3237	CN	EHA101 and EHA105	24	18:6	3	Selection with ammonium glufosinate	3.0%.	[173]
Bert	CN 5–7 d	LBA4404 and EHA105	25	0:24	5	Addition of sodium thiosulfate, cysteine and DTT to the co-culture media	16.4%	[174]
Bert, Harosoy, Jack, Peking, Thorne, Williams 79 and 82, Clark, Essex and Ogden	CN 5–6 d	EHA101	24	24:0 0:24	3 or 5	Including cysteine and DTT during cocultivation increased the transformation efficiency.	8.3%	[168]
Thorne, Williams 79 and 82	HS	EHA101	24	16:8	5	Injured explants in the CN may present oxidative stress reducing TF	3.8%	[169]
Hefeng 25, Dongnong 42, Heinong 37, Jilin 39, and Jiyu 58	CN	EHA105	19–28	0:24	5	Silwet L-77 (0.02%), cysteine (600 mg/L) and low temperature during the co-culture increased TF	11%	[175]
Tianlong 1, Yuechun 03–3 and 04–5	HS	EHA101	23	0:24	3–5	Similar TF among cultivars	4.5%	[176]
Thorne, Williams 79 and 82	HS	EHA101	24	16:8	3–5	Low TF but reproducible, it does not require specific technical manipulation. The use of bar gene decreased the number of chimeric plants.	5.0%	[177]
Pk 416, Js 90–41, Hara-soy, Co1 and Co 2	CN 7 d	LBA4404, EHA101, and EHA105	27	0:24	5	Explants micro-wounded by sonication and vacuum application of vacuum	18.6%	[178]
Kwangan	HS	EHA105 OD: 0.6–0.8	24	18:6	5	Explants wounded with a scalpel and then sonicated for 20 s, additionally a vacuum was applied for 30 s	3.0%	[179]
Heihe 19 and 25, Heinong 37, Ha 03–3, YC-1, YC-2, Zhonghuang 39	HS	EHA101 OD: 0.8–1.0	23	16:8	4	Cocultivation at 233 °C °C for 4 days shows better TF with the addition of silver nitrate and lipoic acid	14.7%	[180]
Tianlong 1, Jack, Purple, DLH, NN419, Williams 82, HZM, NN34, and NN88–1	HS	EHA101 OD: 0.6- 1	23	18:6	3–5	Addition of cysteine to de co culture media increased TF to 36%	7–10%	[181]
Maverick	HS	EHA101 and EHA105	24	16:8	5	Explants exposed to dehydration show better TF. Strain EHA105 shows higher TF	18.7%	[182]
Jack	HS	EHA105 O.D: 0.6	22	0:24	3–5	Co-culture under dehydrating conditions can increase the acceptance of *Agrobacterium* T-DNA	22.9%	[163]
JS335	HS	EHA105 OD: 0.6- 1.0	27	0:24	3	Application of sonication and vacuum for 10 min results in higher TF	38.0%	[183]

CN: Cotyledonary node, HS: Half-seed; Tm: Temperature, Pp: Photoperiod (light: darkness), TF: Transformation frequency, DTT: Dithiotreitol.

### 5.2. Biolistic-Mediated Transformation

Biolistic delivery is a direct genetic transformation technique that uses physical methods to introduce genetic material into plant tissues. The process involves coating gold or tungsten particles with the DNA of interest, preparing the target plant tissues, loading the particles into a gene gun, and subsequently propel them into the explants using high-pressure gas in a biolistic chamber [184]. The impact of these particles on the cell membrane causes temporary rupture, facilitating the entry of DNA into the cell. After bombardment, the explants are placed in culture media to allow recovery and development of the explant. One of the main advantages of biolistic transformation is its applicability across a broad range of plant species, including those that are recalcitrant to *Agrobacterium*-mediated transformation, which often relies on specific plant–pathogen interactions for success [185].

### 5.3. Alternative Transformation Techniques

Although *Agrobacterium*- and biolistic-mediated transformation are the predominant methods for generating transgenic soybean plants, further strategies have been evaluated [75]. These include techniques such as electroporation, where electrical pulses are used to introduce DNA into protoplasts or calli, and polyethylene glycol (PEG)-mediated transformation, which facilitates DNA uptake in protoplast, although the limiting factor is whether the species could regenerate from these cells. Another method, silicon carbide whisker-mediated transformation, uses DNA-coated whiskers (microfibers or microscopic needle-like particles) to penetrate cell walls and deliver genetic material [186,187,188].

In addition, in planta transformation techniques have been also reported, offering a promising alternative for the generation of genetically modified soybean plants. These in planta methods bypass the need for in vitro culture and regeneration stages, although their widespread use remains limited [189,190,191,192].

Additionally, methods such as liposome-mediated transfection, fiber-mediated DNA delivery, laser-induced DNA delivery, and pollen transformation, have not yet been implemented in soybeans, but represent an underexplored toolbox for the genetic improvement of the crop [193].

## 6. Challenges and Perspectives in Developing Stress-Resistant Soybean Cultivars

The current impact of biotic stress on soybean production necessitates further research to elucidate pest and disease interactions. Advanced molecular techniques, such as next generation sequencing, marker-assisted selection, and gene editing, show promise in developing disease-resistant and pest-tolerant soybean varieties. Integrating data-driven approaches and predictive modeling can contribute to forecast disease outbreaks and pest infestations, facilitating timely and effective interventions. Despite these advancements, several challenges remain in regulatory compliance, consumer acceptance, and the need for ongoing research to enhance stress-resistant traits. Public communication and awareness are essential for building trust and integrating genetically modified soybeans into mainstream agriculture practices [194]. In this section, we describe several underexplored elements that we believe hold great potential for improving soybean crop productivity, including microbiome, polyploidy, and epigenetic memory.

### 6.1. Microbiome

The plant microbiome, particularly the rhizosphere microbial communities, plays a crucial role in soybean growth and health by enhancing nutrient uptake, suppressing diseases, and increasing resistance to stresses [195]. The diversity and composition of these microbial communities are influenced by factors such as climate, soil type, plant species, genotype, and growth stage. In soybeans, the rhizosphere is notably enriched with arbuscular mycorrhizal fungi and symbiotic rhizobia, such as *Bradyrhizobium* spp., and *Bacillus* spp. [196]. The symbiotic relationship between *Bradyrhizobium* spp. and the soybean is essential for nitrogen fixation, a critical process for plant development. These symbiotic interactions trigger the differentiation of the plant root and the bacteria into nodules where this process takes place; this process, in turn, results in the synthesis of metabolites beneficial to the plant and the microorganisms [196,197]. The root secretes flavonoids that attract rhizobacteria, which respond by migrating to the root through an infection thread and secreting lipochitooligosaccharides that trigger the formation of structures that facilitate entry of the bacteria into the plant cortex. Upon differentiation of rhizobia into specialized bacteroid cells, activation of genes involved in nodulation occurs where atmospheric nitrogen is fixed into ammonium, which plants can assimilate as glutamate. In return, bacteria receive carbohydrates like malate as an energy source [198,199,200]. This interaction improves plant yield and soil fertility, reducing the need for synthetic fertilizers [201].

Fungal communities in the rhizosphere, primarily from the phyla *Ascomycota* and *Basidiomycota*, are relatively stable but are influenced by agricultural practices such as fertilizer application and rhizobium inoculation. For instance, high-yield field soils are rich in beneficial fungi like *Trichoderma* and *Metarhizium*—which play roles as mycoparasites and entomopathogens—along with *Bradyrhizobium elkanii* and *Flavobacterium*, symbiotic nitrogen fixers and plant-growth promoters. Furthermore, *Corynespora cassiicola*, commonly found in roots from high-yield fields, may have a dual role as a pathogen and a fungal endophyte. In contrast, low-yield fields are dominated by pathogenic fungi such as *Fusarium*, *Macrophomina*, and free-living nitrogen-fixing bacteria like *Klebsiella* and *Kosakonia*, indicating less efficient symbiotic nitrogen fixation. Low-yield soils also show higher abundances of potentially parasitic *Allorhizobium* strains and *Pseudomonas* species (e.g., *P. chlororaphis* and *P. frederiksbergensis*), which exhibit both pathogenic and beneficial properties [202].

The biological processes associated with the soybean microbiome have garnered interest due to their potential to enhance crop yield, thereby reducing the reliance on agrochemicals and synthetic fertilizers. Inoculating soybean crops with nitrogen-fixing bacteria not only improves yield but also helps restore soil fertility by increasing the availability of nitrogen. Integrating the use of endophytic organisms like *Bradyrhizobium* spp. into soybean cultivation strategies is crucial for boosting production and enhancing plant defense against stress conditions [196].

### 6.2. Polyploidy

Polyploidy, characterized by the presence of multiple sets of chromosomes in an organism, is a prevalent evolutionary trait in plants, especially in angiosperms. The soybean has undergone lineage-specific whole-genome duplication events, before and after its divergence from the common bean ancestor, maintaining subgenomic stability during polyploidization and subsequent slow diploidization. As a result of this slow diploidization process, which occurred more than 10 million years ago, approximately 75% of the genes in the soybean genome are present in multiple copies [203].

Ancient and recent polyploidization events have contributed to the enhancement of agronomic traits, such as seed size, pod number, and oil content [204]. Polyploidy in the soybean is associated with increased yield potential, improved stress tolerance, and diversification of metabolic pathways, which collectively bolster the crop adaptability to environmental stresses like salinity and drought [205]. Harnessing polyploidy in soybean breeding offers significant potential for developing high-yielding cultivars with enhanced resilience to biotic and abiotic stresses. Advanced genomic tools, such as high-throughput sequencing and genome-wide association studies, are accelerating the identification of key genes associated with desirable traits in polyploid soybeans [206]. However, challenges remain in fully exploiting the polyploidy potential due to the complexity of its genetic and epigenetic regulation. Understanding the interactions between polyploidy and environmental factors is essential for developing soybean cultivars that can thrive in diverse agroecological settings. The continued integration of polyploidy into breeding programs promises the development of superior soybean cultivars with improved adaptability, nutritional quality, and sustainability meeting the demands of future agricultural production.

### 6.3. Epigenetics and Memory

Epigenetic modifications, such as DNA methylation and histone modifications play a crucial role in regulating stress memory in plants. These modifications enable plants to enhance their adaptative response to recurrent stress by inducing physiological and morphological changes that can persist for several generations (although how many are still debated) contributing to phenotypic plasticity [83]. This plasticity has been taken advantage of for crop improvement, including soybean cultivation. Recent advances in sequencing and genomic technologies have facilitated the identification and characterization of epigenetic changes that occur in response to environmental changes, providing insights into their regulatory mechanisms and impact on plant development [207]. Current methods to induce beneficial epigenetic variations (histone modifications, DNA methylation, and production of small-interfering RNAs) include RNA interference, CRISPR/Cas systems, transcription activation-like effectors (TALEs), and chemical treatments with compounds like azacytidine or zebularine [207,208,209]. These epigenetically modified lines may provide advantages in terms of productivity or increased response to biotic and abiotic stress for some generations, particularly in contexts where the use of transgenic organisms is restricted and can be integrated into traditional breeding programs to enhance genetic diversity and improve crop resilience.

## 7. Concluding Remarks

The economic significance of soybeans remains unparalleled, given their pivotal role in global trade, industrial applications, and human nutrition. As the global demand for sustainable and plant-based alternatives continues to rise, the strategic harnessing of soybean potential is crucial for fostering economic prosperity, environmental sustainability, and societal well-being. Continuous investment in research, technology, and market diversification is imperative to fully unlock the economic potential of soybean cultivation, ensuring its ongoing contribution to global economic development and human welfare. Soybean genetic transformation exemplifies the transformative potential of biotechnology in revolutionizing agriculture. This innovation addresses key challenges in food security and sustainability while offering a promising future for global agriculture. As research and development in this field progress, continued collaboration between scientists, policymakers, and stakeholders is critical to harness the full potential of soybean genetic transformation for the benefit of society and the environment.

Genetic transformation has enabled the development of soybean cultivars with enhanced resistance to pests and diseases, thereby reducing the reliance on chemical pesticides. The incorporation of genes conferring tolerance to specific herbicides has facilitated effective weed management, leading to improved crop yields and reduced production costs. Furthermore, genetic modifications to enhance nutritional quality have resulted in soybean cultivars with higher protein content, altered fatty acid profiles, and reduced levels of antinutritional factors, thereby improving human and animal nutrition. Despite the remarkable advancements, challenges remain such as regulatory constraints, public acceptance, and potential environmental impacts. Balancing innovation and biosafety are critical for the widespread adoption of genetically modified soybean cultivars. Further research is required to enhance the efficiency of genetic transformation techniques and to explore the full potential of the soybean genome to address evolving agricultural and nutritional needs.

The future of soybean agriculture hinges on ongoing research and development to improve important traits such as drought tolerance, pest and disease resistance, as well as better seed quality. Efforts to combine these traits into a single cultivar are expected to produce more resilient and productive soybean cultivars. Additionally, as climate change continues to impact agriculture, developing new soybean cultivars that can thrive in shifting environmental conditions is imperative.

## Figures and Tables

**Figure 1 plants-13-03073-f001:**
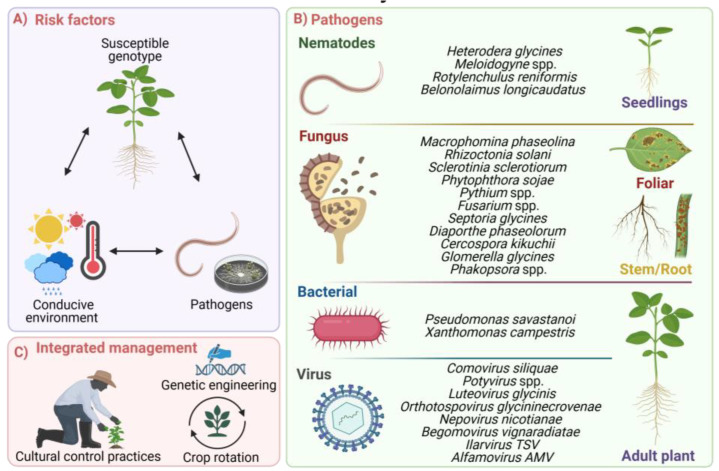
Biotic stress in soybean cultivars. (**A**) Incidence of pathogens and diseases depends on environmental conditions, the susceptibility of the soybean cultivars, and pathogen adaptation. (**B**) Pathogens cause the most important economic losses in soybeans. (**C**) Integrated management actions to mitigate the most common diseases.

**Figure 2 plants-13-03073-f002:**
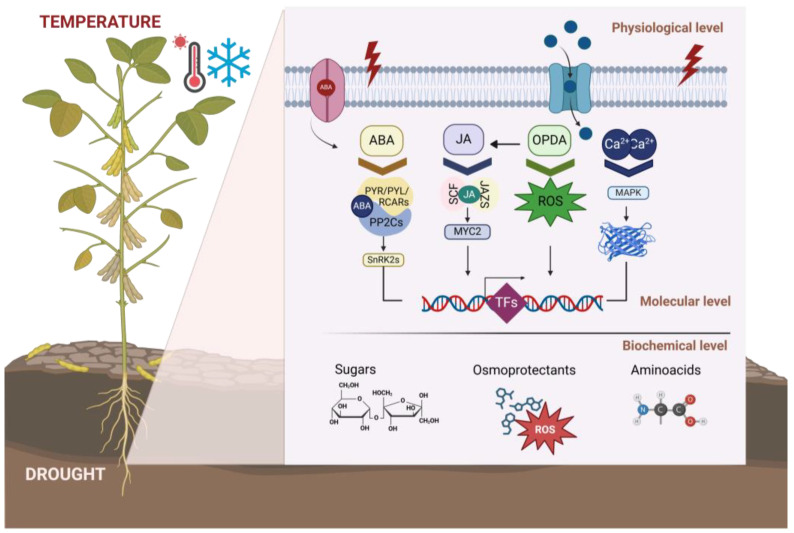
Response to drought stress in soybean germplasm. Signaling pathways to cope with drought stress conditions at the molecular and biochemical levels. Limited water availability and extreme temperatures induce ABA-dependent or independent pathways, including the synthesis of water channels, the expression of signal transduction molecules, and transcription factors that regulate such pathways. At the biochemical level, this type of stress activates the accumulation of osmoprotectant sugars, modified amino acids, and anti-oxidative enzymes.

**Figure 3 plants-13-03073-f003:**
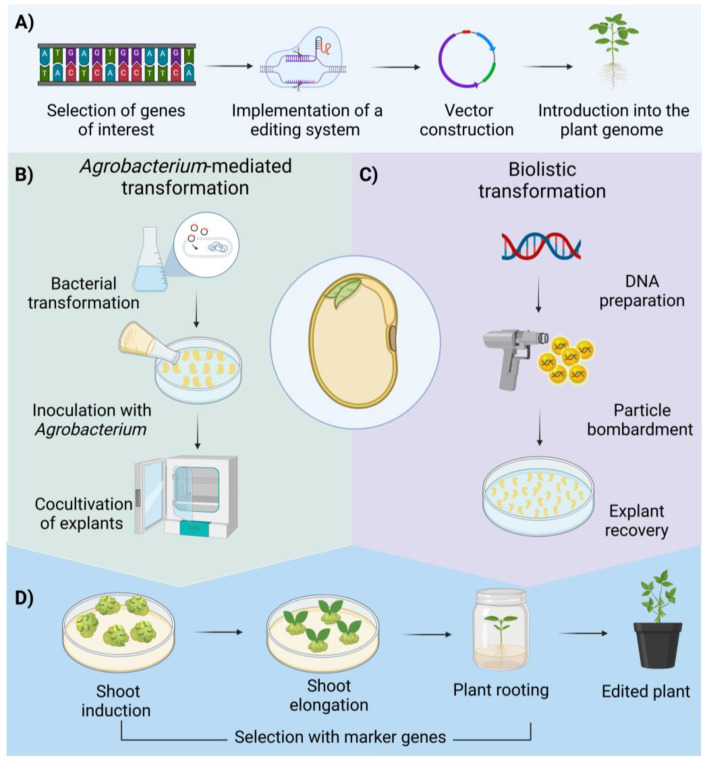
Process of soybean genetic transformation. (**A**) Isolation and manipulation of specific genes. (**B**) Introduction into the plant genome using *Agrobacterium tumefaciens.* (**C**) Biolistic mediated transformation. (**D**) Regeneration and selection of transformed explants.

**Table 1 plants-13-03073-t001:** Examples of gene editing in soybean and associated phenotypes.

Category	Gene Name	Gene Function	Phenotype	Reference
Abiotic stress	*GmHdz4*	Homeodomain-leucine zipper TF	Enhanced drought tolerance, increased root length, area, and number of root tips	[105]
	*GmAITR*	ABA-induced transcription repressor	Enhanced salinity tolerance	[106]
	*GmPLA*	Phospholipase A	Flooding and drought tolerance. Modified response in iron and phosphorus deficiency.	[107]
	*GmWRKY46*	*WRKY* TF family induced by phosphorous deprivation in roots.	Enhanced tolerance in phosphorous deficiency	[108]
	*GmPYL*	Receptor involved in ABA signal transduction.	Reduction in ABA sensitivity. Higher seed germination rate, plant height and branch number.	[109]
	*GmLHY*	TF that regulates the ABA signaling pathway	Improved drought tolerance	[110]
Architecture, plant morphology, and production	*GmDWF1*	Synthesis of brassinosteroids	Increase in pod production	[111]
	*GmPDH1*	Synthesis and distribution of lignin in the pod internal sclerenchyma.	Pod shattering resistance	[112]
	*GmLHY*	MYB TF involved in the gibberellic acid pathway	Reduction in plant height and shortened internodes	[113]
	*GmSPL9*	TF involved in regulating plant architecture	Shorter plastochron length and increased node number	[114]
	*GmJAG*	Homolog of JAGGED TF in *Arabidopsis*	Increase in number of seeds per pot and higher yield	[115]
	*GmE4*	Homolog of phytochrome A in *Arabidopsis*	Earlier maturation time	[116]
	*GmCPR5*	Regulation of the plant immune response, cell cycle, and development	Short trichomes with smaller nuclei	[117]
	*GmEOD* and *GmAIP2*	E3 ubiquitin ligases	Larger seed size with higher protein and oil content	[118,119]
	*GmmiR396*	Negative regulator of grain size	Enlarged seed size and increased yield	[120]
	*GmKIX8*	Negatively transcriptional regulator of cell proliferation	Increased seeds and leaves size	[121]
Biotic stress	*GmF3H and GmFNSII-1*	Flavanone-3-hydroxylase and flavone synthase II	Increase in isoflavone content and resistance to SMV	[122]
	*GmUGT*	UDP-glycosyltransferase	Resistance to leaf-chewing insects	[123]
	*GmCDPK38*	Calcium-dependent protein kinase	Increased resistance to *Spodoptera litura* and late flowering	[124]
	*GmTAP1*	Histone acetyltransferase	Enhanced resistance to *Phytophthora sojae*	[125]
	*GmARM*	ABA-related signaling pathway protein	Salt tolerance, alkali and *Phytophthora sojae* resistance	[126]
	*GmSNAP02*	Vesicle fusion in intracellular trafficking	Resistance to soybean cyst nematode	[127]
Flowering time	*GmEIL* and *GmEIN*	Ethylene signal perception and transduction	Early flowering and increased yield	[128]
	*GmE1*, *GmFKF*, *GmTOE4b*, and *GmLNK*	TF and regulators of GmFT2a y GmFT5a	Early flowering	[129,130,131,132,133,134]
	*GmFT*, *GmAP1*, *GmNMHC5*, and *GmSOC1*	Florigen protein and proteins related to identity of floral organs	Adaptation for planting in low latitudes (late flowering)	[135,136,137,138,139]
Herbicide resistance	*GmAHAS*	Biosynthesis of branched-chain amino acids	Resistance to five AHAS-inhibiting herbicides	[140]
Male sterility	*GmMS1*	Sporophytic controlling factor for anther and pollen development	Male sterility (useful trait for breeding applications)	[141,142]
	*GmAMS*	bHLH TF that affects tapetal development		[143]
Nodulation	*GmRIC*	Nodule-enhanced CLE peptide	Increase in nodule number per plant	[144,145]
Nutritional and organoleptic value	β-conglycinin (7S) and glycinin (11S)	Storage proteins	Changes in emulsion stability and gelling ability in soy protein	[146]
	*GmlincCG1*	Long noncoding RNA mapped to the intergenic region of the 7S α-subunit locus	Deficiency of the allergenic α’, α, and β subunits of 7S	[147]
	*GmRS*, *GmSTS*, and *GmGOLS*	Raffinose and stachyose biosynthesis	Reduction in anti-nutritive oligosaccharides	[80,148]
	*GmIPK1* and *GmMRP5*	Synthesis and transport of phytic acid	Reduced phytic acid content in seeds	[149,150]
	*GmLOX*	Lipoxygenases related with “beany” flavor	Beany flavor reduction	[151]
	*GmKTI* and *GmBBi*	Kunitz and Bowman–Birk protease inhibitors	Low trypsin and chemotrypsin inhibitor content	[152,153]
	*GmBADH2*	Aminoaldehyde dehydrogenase, involved in the formation of 2-AP	Confers a “pandan-like” (high value quality trait) aroma	[154]
Oil profile	*GmFAD2*	Conversion of oleic acid to linoleic acid	High oleic, low linoleic, and alfa-linoleic phenotype	[155,156,157,158,159]
	*GmPDCT*	Phosphatidylcholine: diacylglycerol cholinephosphotransferase		[160]

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
