# Peer review of "Advances in Soybean Genetic Improvement"

_plants, 2024, doi:10.3390/plants13213073_

Round 1
Reviewer 1 Report
Comments and Suggestions for Authors
This review is exceptionally well written and informative. Extensive references are provided for the reader's benefit.
Minor editorial suggestions are given below.
Throughout the manuscript the authors frequently use the word 'variety' but at other times 'cultivar'. Search and replace. Use cultivar throughout.
Line 36, insert 'global' before commercial
Line 39, insert (on a 13% moisture basis) after 35% protein
Line 56, change heat to day length
Lines 70-73; 87-95; 103-106 add the common name to be consistent with above.
Line 139, insert 'and' between xylem and translocated
Line 140, tomato - insert Genus, species
Line 158, common bean - insert Genus species
Figure 1, under Virus, give the Latin to be consistent with above
Line 245, check the references. I believe Glycine max Plant Introductions were used, not wild species
Line 261-263 Mention herbicide resistance as the #1 focus of genetic engineering in soybean for commercialization purposes.
Line 309. Delete 'precise'. Transformation is not precise. CRISPR Cas-9 gene editing can be precise.
Line 318, delete 'gene gun bombardment' as it is redundant to biolistic particle delivery systems.
Figure 3. Place Figure Caption at the bottom of figure
Note beginning at Section 7, manuscript line numbering is not continuous and re-starts with the number 1.
Line 26 (Section 8), switch positions of enhances digestibility and reduces allergenic potential to describe protein inhibitors and phytic acid respectively.
Section 9.2, Line 88 change 'antecessor' to 'ancestor'
Section 10, Line 117, consider alternative word choice for 'stands for'
No further edits.
Excellent paper!
Author Response
Reviewer 1
This review is exceptionally well written and informative. Extensive references are provided for the reader's benefit. Minor editorial suggestions are given below.
Throughout the manuscript the authors frequently use the word 'variety' but at other times 'cultivar'. Search and replace. Use cultivar throughout.
Thank you very much for your observation. The word “variety” was replaced with “cultivar” throughout the text.
Line 36, insert 'global' before commercial
We made the suggested change.
Line 40-41: …“In 2020, the USA, Brazil, and Argentina contributed to 80% of global soybean production with an estimated of 338 million metric tons [4–6]”…
Line 39, insert (on a 13% moisture basis) after 35% protein.
Thank you for your observation. We include the information provided
Line 48-50: …“With 35% protein content (on a 13% moisture basis), soybean surpasses other staple crops like wheat, rice, and corn, and is rich in antioxidants, minerals, vitamins, and dietary fiber [13,14]”…
Line 56, change heat to day length
We made the suggested change.
Line 64-66: …“Soybean plants typically take 75-80 days to reach full maturity, depending on day length exposure”…
Lines 70-73; 87-95; 103-106 add the common name to be consistent with above.
Thank you for your observation. To address your valuable comment and to maintain consistency throughout the Introduction we have decided to include only the binomial scientific names of each organism in this section.
Line 139, insert 'and' between xylem and translocated
We made the suggested change.
Line 145-148: …“ABA is synthesized in roots and likely in the leaf vascular system; it has been postulated that ABA is transported from the root to leaves and in general aerial tissues through the xylem and translocated where it induces stomatal closure [43]”…
Line 140, tomato - insert Genus, species
We made the suggested change.
Line 148-150: …“However, there is evidence that ABA impacts more stomatal opening than the xylem, at least in tomato (Solanum lycopersicum) [44]”…
Line 158, common bean - insert Genus species
We made the suggested change.
Line 168-171: …“Some mRNAs induced by water deficit in a common bean (Phaseolus vulgaris) cultivar accumulate in the phloem, suggesting that such tolerance involves long-distance signaling through the vasculature, and may be a more general phenomenon [39].”
Figure 1, under Virus, give the Latin to be consistent with above
Thank you for your observation. We have corrected the figure and expanded the information presented.
Line 245, check the references. I believe Glycine max Plant Introductions were used, not wild species.
Thank you for your observation. We have modified the text according to your suggestion.
Line 254-257: “Comprehensive analysis of commercial soybean plant introductions has identified SNPs as key elements for improving traits such as protein and oil content, as well as enhancing tolerance to biotic and abiotic stress [86]”.
Line 261-263 Mention herbicide resistance as the #1 focus of genetic engineering in soybean for commercialization purposes.
Thank you for your suggestion. We included the information as indicated.
Line 264-266: …“In addition, herbicide resistance, remains the primary focus of genetic engineering in soybean for commercialization purposes, given its significant role in facilitating large-scale agricultural practices”…
Line 309. Delete 'precise'. Transformation is not precise. CRISPR Cas-9 gene editing can be precise.
We totally agree with your observation, we made the suggested change.
Line 337-338: “Genetic transformation is a biotechnological approach enabling modification of the plant genome to incorporate desirable traits [75,99]”…
Line 318, delete 'gene gun bombardment' as it is redundant to biolistic particle delivery systems.
We agree with this observation. We deleted 'gene gun bombardment'.
Line 345-347: …“The selected DNA sequences are introduced into the soybean genome using methods such as Agrobacterium-mediated transformation, gene gun bombardment, or biolistic particle delivery systems [101]”…
Figure 3. Place Figure Caption at the bottom of figure. Note beginning at Section 7, manuscript line numbering is not continuous and re-starts with the number 1.
Thank you for the observation. The suggested changes have been made.
Line 26 (Section 8), switch positions of enhances digestibility and reduces allergenic potential to describe protein inhibitors and phytic acid respectively.
Thanks for your observation. We switched the words according to your recommendation.
Line 422-425: …“Improvements in soybean seed quality also focus on reducing antinutritional factors, such as protein inhibitors and phytic acid, which reduces allergenic potential and enhances digestibility”…
Section 9.2, Line 88 change 'antecessor' to 'ancestor'
Thank you for your observation. We made the suggested change.
Line 486-489: …“Soybean has undergone lineage-specific whole-genome duplication events, before and after its divergence from common bean ancestor, maintaining subgenomic stability during polyploidization and subsequent slow diploidization”…
Section 10, Line 117, consider alternative word choice for 'stands for'
Thank you for your comment. We replaced 'stands for' with 'exemplifies'.
Line511-512: “Soybean genetic transformation exemplifies the transformative potential of biotechnology in revolutionizing agriculture.”
No further edits.
Excellent paper!
We thank you very much for your valuable and professional revision that substantially improved the manuscript.
Reviewer 2 Report
Comments and Suggestions for Authors
soybeans hold global significance as a major source of nutrition for both humans and animals. This review has carried on the elaboration to many aspects. It is useful for soybeans variety improvement and new germplasm creation. However, this manuscript still has some deficiencies, please making modifications and improvements in the following aspects.
1. The title is too big, and the title is not entirely appropriate for content of the manuscript.
2. The abstract needs to be rewritten, especially lines 13-22. Abstract can not be written as a Introduction, it is important to summarize the methods, content, conclusions and so on.
3. The content of the review has too many aspects, and the analysis is not deep enough. The content should be focused on the analysis of a certain aspect of the content, as far as possible in-depth analysis. For example, the content of 4, 7. 8. 9. 10 are too many, The analysis is shallow and of little significance.
Author Response
Comments and Suggestions for Authors
soybeans hold global significance as a major source of nutrition for both humans and animals. This review has carried on the elaboration to many aspects. It is useful for soybeans variety improvement and new germplasm creation. However, this manuscript still has some deficiencies, please making modifications and improvements in the following aspects.
- The title is too big, and the title is not entirely appropriate for content of the manuscript.
Following your suggestions, we changed the title to: 'Advances in Soybean Genetic Improvement'.
- The abstract needs to be rewritten, especially lines 13-22. Abstract can not be written as a Introduction, it is important to summarize the methods, content, conclusions and so on.
We agree with your assertion regarding the abstract. Therefore, we have decided to rewrite the Abstract as indicated below:
Line 12-27: “Abstract: Soybean (Glycine max) is a globally important crop due to its high protein and oil content, serving as a key resource for human and animal nutrition, as well as bioenergy production. This review assesses recent advancements in soybean genetic improvement by conducting an extensive literature analysis focusing on enhancing resistance to biotic and abiotic stresses, improving nutritional profiles, and optimizing yield. We also describe the progress in breeding techniques, including traditional approaches, marker-assisted selection, and biotechnological innovations such as genetic engineering and genome editing. Key advancements include the development of transgenic soybean cultivars through Agrobacterium-mediated transformation and biolistic methods aimed at introducing traits such as herbicide resistance, pest tolerance, and improved oil composition. However, challenges remain, particularly with respect to genotype recalcitrance to transformation, plant regeneration, and regulatory hurdles. In addition, we examined how wild soybean germplasm and polyploidy contribute to expanding genetic diversity as well as the influence of epigenetic processes and microbiome on stress tolerance. These genetic innovations are crucial for addressing the increasing global demand for soybeans, while mitigating the effects of climate change and environmental stressors. The integration of molecular breeding strategies with sustainable agricultural practices offers a pathway for developing more resilient and productive soybean varieties, thereby contributing to global food security and agricultural sustainability.”
- The content of the review has too many aspects, and the analysis is not deep enough. The content should be focused on the analysis of a certain aspect of the content, as far as possible in-depth analysis. For example, the content of 4, 7. 8. 9. 10 are too many, The analysis is shallow and of little significance.
Thank you very much for your feedback. We agree with your observations regarding the content and the lack of depth in certain sections. In response, we have revised the manuscript by focusing on fewer aspects and providing a more in-depth analysis. Specifically, we have addressed the sections you mentioned and made the necessary adjustments to improve the quality and relevance of the review.
Reviewer 3 Report
Comments and Suggestions for Authors
Interesting review in a very interesting species for the agricultural world. The review after taking into account biotic and abiotic stresses comprehensively describes different methods of genetic transformation of soybean.
I recommend adding the chromosome number (complete formula) and the reproductive system of the species in the introduction.
Line 111: change interactions between in interactions among
Line 169: Figure 1 even if it is understood well, I still recommend adding the letters shown in the caption to the figure
Author Response
Interesting review in a very interesting species for the agricultural world. The review after taking into account biotic and abiotic stresses comprehensively describes different methods of genetic transformation of soybean.
I recommend adding the chromosome number (complete formula) and the reproductive system of the species in the introduction.
Thank you very much for your valuable comment. We included the requested information in the Introduction section.
Line 34-38: …“The species has 40 chromosomes (2n = 40), and its reproductive system is predominantly self-pollinating, with cleistogamous flowers, leading to a 97-99% self-pollination rate [2,3]”…
Line 111: change interactions between in interactions among
Thank you for your suggestion. We made the requested change.
Line 120-121: “The complex interactions among biotic stressors demand the development of comprehensive management strategies [31]”…
Line 169: Figure 1 even if it is understood well, I still recommend adding the letters shown in the caption to the figure
Following your suggestions, we incorporated the corresponding letters to the caption figure.
We thank you very much for your valuable and professional revision that substantially improved the manuscript.
Round 2
Reviewer 2 Report
Comments and Suggestions for Authors
The current manuscript has made some progress compared to before. But it did not reach the ideal state.